# Old and New Systemic Immune-Inflammation Indexes Are Associated with Overall Survival of Glioblastoma Patients Treated with Radio-Chemotherapy

**DOI:** 10.3390/genes13061054

**Published:** 2022-06-13

**Authors:** Francesco Pasqualetti, Celeste Giampietro, Nicola Montemurro, Noemi Giannini, Giovanni Gadducci, Paola Orlandi, Eleonora Natali, Paolo Chiarugi, Alessandra Gonnelli, Martina Cantarella, Cristian Scatena, Giuseppe Nicolò Fanelli, Antonio Giuseppe Naccarato, Paolo Perrini, Gaetano Liberti, Riccardo Morganti, Maria Franzini, Aldo Paolicchi, Giovanni Pellegrini, Guido Bocci, Fabiola Paiar

**Affiliations:** 1Radiation Oncology Unit, Azienda Ospedaliero Universitaria Pisana, 56124 Pisa, Italy; noemi.giannini@yahoo.com (N.G.); ggadducci2@gmail.com (G.G.); gonnelli.alessandra@gmail.com (A.G.); martina.cantarella@ao-pisa.toscana.it (M.C.); fabiola.paiar@unipi.it (F.P.); 2Department of Oncology, University of Oxford, Oxford OX1 4BH, UK; 3UO Laboratorio Analisi Chimico Cliniche, Azienda Ospedaliero Universitaria Pisana, 56124 Pisa, Italy; c.giampietro@ao-pisa.toscana.it (C.G.); eleonoranat93@gmail.com (E.N.); p.chiarugi@ao-pisa.toscana.it (P.C.); gn.pellegrini@ao-pisa.toscana.it (G.P.); 4Neurosurgery Unit, Azienda Ospedaliero Universitaria Pisana, 56124 Pisa, Italy; nicola.montemurro@unipi.it (N.M.); paolo.perrini@unipi.it (P.P.); g.liberti@ao-pisa.toscana.it (G.L.); 5Department of Clinical and Experimental Medicine, University of Pisa, 56126 Pisa, Italy; paolaorlandi21@libero.it (P.O.); guido.bocci@unipi.it (G.B.); 6Department of Translational Research and New Technologies in Medicine and Surgery, University of Pisa, 56126 Pisa, Italy; cristian.scatena@unipi.it (C.S.); gnfanelli@gmail.com (G.N.F.); giuseppe.naccarato@med.unipi.it (A.G.N.); maria.franzini@unipi.it (M.F.); aldo.paolicchi@unipi.it (A.P.); 7Department of Statistics, University of Pisa, 56126 Pisa, Italy; r.morganti@ao-pisa.toscana.it

**Keywords:** systemic immune-inflammation index, glioblastoma, SII, NPW/LM, NPM/L, inflammation index

## Abstract

**Background**. Systemic immunity and inflammation indexes (SI) derived from blood cells have gained increasing attention in clinical oncology as potential biomarkers that are associated with survival. **Materials and methods**. We tested 12 different SI using blood tests from patients with isocitrate dehydrogenase 1 and 2 wild-type glioblastomas, treated with radio-chemotherapy. The primary endpoint was their overall survival. **Results**. A total of 77 patients, comprising 43 males and 34 females, with a median age of 64 years (age range 26–84), who were treated between October 2010 and July 2020, were included in the present analysis (approved by a local ethics committee). In the univariate Cox regression analysis, all the indexes except two showed a statistically significant impact on OS. In the multivariate Cox regression analysis, neutrophil × platelet × leukocyte/(lymphocyte × monocyte) (NPW/LM) and neutrophil × platelet × monocyte/lymphocyte (NPM/L) maintained their statistically significant impact value. **Conclusions**. This univariate analysis confirms the potential of systemic inflammation indexes in patients with glioblastoma, while the multivariate analysis verifies the prognostic value of NPW/LM and NPM/L.

## 1. Introduction

Glioblastoma (GBM) is the most common and most aggressive malignant primary brain tumor in adults, resulting in dire prognoses and mortality rates overlapping the incidence [1]. Although several treatments have been proposed, radio-chemotherapy (RT-CT) with temozolomide has represented the standard of care since 2005; unfortunately, the survival of GBM patients has not significantly improved over the previous decades [2,3,4].

To overcome the limits reached in GBM therapy, one of the main issues to be addressed in neuro-oncology is the lack of biomarkers necessary for planning translational studies. While previous attempts to identify novel radio- and chemo-sensitive biomarkers in GBM have occurred, they were focused on tumor samples and experienced a series of limitations, such as the difficulty of obtaining tumor tissue from all GBM patients [5,6,7,8,9].

As the SI derived from blood cells may potentially integrate the impact of biomarkers derived from tumor tissue, in recent years, they have gained increasing attention in the field of clinical oncology [10,11,12]. Hence, various immune/inflammation blood products have been investigated to create novel prognostic or predictive models in several solid cancers. Within this context, an elevated neutrophil-to-lymphocyte ratio (NLR) or platelet-to-lymphocyte ratio (PLR) has been associated with increased aggressiveness and shorter survival times in numerous solid cancers [13,14]. Recently, a novel index called the systemic immune-inflammation index (SII), which is based on neutrophil (N), platelet (P), and lymphocyte (L) counts, has emerged and reflects comprehensively the balance of host inflammatory and immune statuses [15]. Moreover, literature limited to a few, and very heterogeneous, studies also suggest that the different NLR, PLR, or SII may be associated with the different biology and overall survival in glioblastoma patients [11]. Recently, the dynamic changes in the systemic immune-inflammation index have been associated with the prediction of the prognoses of patients with brain metastases of lung adenocarcinoma who are treated with radiotherapy [16]. Moreover, the SII has been described as an independent prognostic indicator for the characterization and identification of patients harboring NSCLC brain metastases, who are at high risk of recurrence after radiotherapy [17].

Considering the unmet need for novel biomarkers in neuro-oncology, the easy reproducibility of SI, and their low cost and invasiveness, the present retrospective monocentric study aims at assessing the impact of a series of systemic immunity and SI in a homogenous group of patients who have been diagnosed with glioblastoma. Therefore, the scope of this analysis is to integrate tumor tissue’s prognostic power.

## 2. Materials and Methods

This study represents a retrospective and monocentric study carried out at Pisa University Hospital and approved by the Local Institutional Review Board (Prot. Number 560/2015). All data were retrieved from the Pisa University Hospital dataset. All procedures were conducted according to the ethical standards of the institutional and/or national research committee, and in compliance with the 1964 Helsinki declaration and its later amendments or comparable ethical standards.

## 3. Patients and Systemic Inflammation Indexes

For the present study, we selected patients diagnosed with glioblastoma who were referred to the Pisa University Hospital multidisciplinary team. To make the cohort as homogeneous as possible, we picked out only those patients older than 18 years, with isocitrate dehydrogenase (IDH) 1 and 2 wild-type glioblastomas, a Karnofsky performance status (KPS) at the time of diagnosis of ≥70, available pretreatment complete blood count tests, no history of active immunosuppressive therapies, and who were free of corticosteroid therapy. Methylation of the methyl-guanine-methyltransferase (MGMT) gene promoter was not available for an adequate number of patients, so we decided to exclude this data from the analysis. All patients underwent surgery (gross tumor removal or partial tumor removal (including biopsy)), postoperative radiotherapy (60 Gy/30 fractions, delivered using the volumetric modulated arc therapy technique with concomitant daily temozolomide), and sequential chemotherapy with temozolomide (planned up to 12 cycles). After the diagnosis of disease recurrence, every patient was treated with salvage therapies (second-line chemotherapy, with or without second surgery and/or re-irradiation) or the best supportive care [3,18,19]. The overall survival (OS) rate was calculated from the beginning of radiotherapy until death or the last follow-up. Blood tests performed just before neurosurgery were used to calculate the inflammation indexes. We used the existing inflammatory indexes, such as SII and PLR, alongside novel inflammatory indexes based on blood tests. To calculate SI, the staff at the Department of Clinical Chemistry Laboratory used peripheral blood, drawn into EDTA-K2 tubes using vacuum tube needles. Hematological parameters, such as the total white cell count (W), the differential white cell count (N, L, monocytes M, eosinophils, and basophils), and P count were obtained using a Sysmex XE-2100 (Sysmex, Kobe, Japan) automated blood analyzer and the related reagents, used strictly in accordance with the manufacturer’s instructions.

The normal reference range is 4.0–11.0 1000/mmc for W, 1.80–7.00 1000/mmc for N, 0.90–4.50 1000/mmc for L, 0.1–1.2 1000/mmc for M, and 140–450 1000/mmc for P.

The following formulas were adopted to calculate the inflammation indexes:PLR = P/LPW/L = P × W/LSII = N × P/LNPW/L = N × P × W/LNPM/L = N × P × M/LNPMW/L = N × P × M × W/LNPM/LW = N × P × M/(L × W)NP/LM = N × P/(L × M)NP/(L + M) = N × P/(L + M)NPW/LM = N × P × W/(L × M)NP/WLM = N × P/(W × (L + M))NPW/(L + M) = N × P × W/(L + M)

## 4. Statistical Analysis

Categorical data were described according to absolute and relative frequency, as continuous data with mean and standard deviations. The median, 25th, and 75th percentile values were identified for each index; therefore, overall survival curves were calculated using the Kaplan–Meier method, while the log-rank test was applied to evaluate the differences between curves. Univariate survival analysis of the predictive continuous factors was performed, and the significant factors were subsequently dichotomized by the quartile method. All the calculated categorical factors were subjected to Cox regression as a part of multivariate survival analysis (adjusting for surgery), using a stepwise method; a hazard ratio with a 95% confidence interval was indicated. The primary endpoint was OS. Because not all patients were evaluated using the RANO criteria, we decided not to use progression-free survival as a secondary endpoint (the time of GBM recurrence may vary, when using different criteria to assess the disease progression). Significance was fixed at 0.05 and all analyzes were carried out by SPSS v.28 technology (IBM Corp, released 2021. IBM SPSS Statistics for Windows, Version 28.0. IBM Corp, Armonk, NY, USA).

## 5. Results

Data analysis was performed in October 2021. A total of 77 patients, 43 males and 34 females, with a median age of 64 years (age range 26–84), who were treated between October 2010 and July 2020 were included in the present study. After a median follow-up of 23 months (range 3–94 months), the median OS rate was 17 months (95% CI = 15.3–18.7). The patients’ clinical characteristics are reported in Table 1. 

In the univariate Cox regression analysis, all indexes except NPW/(L + M) and PW/L showed a statistically significant impact on OS rate (Table 2). In the multivariate Cox regression analysis, NPM/L and NPW/LM maintained their statistically significant impact. The median OS rate in patients with NPM/L ≥1500 was 10 months (95% CI = 3.8–16.2 months), whereas in patients with NPM/L <1500, it was 17 months, *p* = 0.014, HR 2.98 (95% CI = 1.243–7.161). The median OS rate in patients with NPW/LM ≥5000 was 11 months, whereas it was 18 months in patients with NPW/LM <5000 (95% CI = 15.5–20), *p* = 0.012, Hazard ratio (HR) 2.411 (95% CI = 1.218–4.771). The data from the multivariate analysis of the two indices remained statistically significant even when assessing the impact of the type of surgery on patient survival. Figure 1 and Figure 2 report the overall survival rate based on NPW/LM and NPM/L value stratification, respectively.

## 6. Discussion

The identification of biomarkers that are easy to find, less expensive to study, and easily reproducible has the potential to enhance the prognostic stratification of glioblastoma patients. Therefore, the recognition of novel prognostic factors in glioblastoma patients represents a clinical challenge in neuro-oncology. Although the underlying mechanisms are not yet fully known, through the study of blood cells we will be able to level up the research into prognostic and predictive biomarkers in patients with glioblastoma, which are currently limited to the analysis of tumor tissue.

From this perspective, the present study reports a monocentric retrospective analysis aimed at studying the impact of systemic inflammation indexes in patients with glioblastoma. To reduce the biases related to the inhomogeneity of patients, we decided to analyze only adult patients with a pathological diagnosis of IDH 1/2 wild-type glioblastoma and a KPS greater than 70, who were treated with post-operative radio-chemotherapy and temozolomide. By excluding IDH1/2-mutated patients, we reduced the risk of enrolling patients suffering from low-grade glioma with different prognoses, microenvironments, and biology [1,20].

Moreover, we considered the OS rate from the first day of radiotherapy, to reduce the impact of different timings between surgery and the start of radiotherapy. The median OS rate reported in this study is consistent with the literature [2]. We decided not to use progression-free survival as a secondary endpoint, to avoid biases related to the use of different disease assessment criteria (not all patients received radio-chemotherapy after introducing RANO criteria in clinical practice [21]). Further studies carried out on selected patients assessed using the same clinical and radiological criteria to establish disease recurrence will be necessary to investigate the role of SI indexes in Progression Free Survival (PFS). Furthermore, because several physiological and pathological elements can modify the number of circulating blood cells, not all patients affected by glioblastoma will be suitable for the analysis of SI. Therefore, to avoid interferences related to surgical procedures or corticosteroid intake, we analyzed only those patients with blood tests obtained before surgery and the administration of corticosteroids. Indeed, when blood samples obtained before the beginning of radiotherapy were analyzed to calculate SI, interestingly, the effects of tissue repair following surgery (which precedes the start of radiotherapy by 4–6 weeks) prevented us from finding any correlation with the OS rate (data not shown).

In the univariate analysis, all the circulating SI assessed, except for NPW/(L + M) and PW/L, showed an impact on overall survival, then NPW/LM and NPM/L also continued to be statistically significant in the multivariate analysis. Patients with an NPW/LM of ≥5000 had a shorter OS rate (11 vs. 18 months, respectively, *p* = 0.012), while patients with an NPM/L of ≥1500 had an even shorter survival rate (10 vs. 17 months, respectively, *p* = 0.014). It is important to note that the data from the multivariate analysis of the two indices remains statistically significant, even when assessing the impact of the type of surgery on patient survival.

All indexes with significant prognostic value for univariate and multivariate analyses had the values of platelets and neutrophils at the numerator. Recently, the role of platelets has been studied in several processes other than hemostasis [22]. For example, the role of platelets has been investigated in the modulation of the immune system and inflammation. Moreover, due to a loop between tumor cells that stimulates platelet production in the hemopoietic marrow through a paracrine way and the platelets themselves, promoting tumor neoangiogenesis and protecting the neoplastic cells from the immune system, the increased number of platelets is supposed to influence cancer progression [23]. Thereafter, an increase in platelet levels contributes to the unfavorable prognoses seen in cancer patients [22]. In patients with a diagnosis of high-grade glioma, an increase in platelets has been observed, compared to blood samples taken 6 months before diagnosis [24]. Furthermore, a preoperative increase in the number of platelets has already been associated with poor prognosis in a series of solid cancers, including glioblastoma [25,26]. In 2020, Marini et al. identified a statistically significant impact on the univariate and multivariate analyses of preoperative platelet counts in 124 glioblastoma patients [27]. Also in the present study, we found that a high platelet value was associated with a worse prognosis. Several patients in this study started prophylactic anticoagulant therapy immediately after surgery. Therefore, it was impossible to assess the correlation between blood platelet levels and thromboembolic events. In addition to the value of the circulating blood platelets, the value of neutrophils may also contribute to the impact of inflammation indexes on OS. It is well established that the main task of neutrophils is related to defense; however, as reported in several studies concerning solid cancers, neutrophil activation plays a leading role in chronic inflammation and tumor progression [28,29]. In our case study, the neutrophil value was present in the indexes of inflammation and was correlated statistically significantly with the OS rate. Our results are in line with a series of clinical studies reporting high blood neutrophil counts in GBM patients, which are associated with a state of immunosuppression, resistance to treatment, and a worse OS rate [14,30,31,32]. Even though the mechanism underlying neutrophilia, the recruitment of neutrophils to the tumor infiltrate, and the interplay between tumor cells and non-tumor cells is not yet well understood, once neutrophils reach the inflamed or damaged site, they are crucial for activating innate and adaptive immunity [32]. By activating a paracrine communication system, the increase of neutrophils in the tumor microenvironment of glioblastoma contributes to reducing the immune response against the tumor and protecting it from medical therapy [8,33]. Moreover, to confirm the link between the number of peritumoral neutrophils and the greater biological aggressiveness of gliomas, patients with IDH-1 wild-type glioma, which is mostly associated with a poorer prognosis, have higher peritumoral neutrophil levels than low-grade, IDH1-mutated gliomas [32,34,35].

To the best of our knowledge, the role of NPM/L and NPW/LM has never been assessed before in patients with glioblastoma, making our results innovative and worthy of further investigation.

The present study evaluated the impact of 12 inflammation indexes in a homogeneous group of glioblastoma patients. Although this study assessed the effect of different inflammation indexes simultaneously for the first time, it still has some limitations. First, the retrospective design and a limited cohort of patients made it impossible to include a few of the clinical parameters related to survival in the multivariate analysis (e.g., MGMT gene methylation and patient age). In addition, the lack of a control group of GBM patients treated without radio-chemotherapy makes it difficult to understand whether the value of these indexes may only be in a prognostic role.

## 7. Conclusions

Our results confirm the potential of SI indexes in patients with glioblastoma. Prospective studies with a larger sample size may lead to the clinical validation of further novel indexes with greater prognostic power.

## Figures and Tables

**Figure 1 genes-13-01054-f001:**
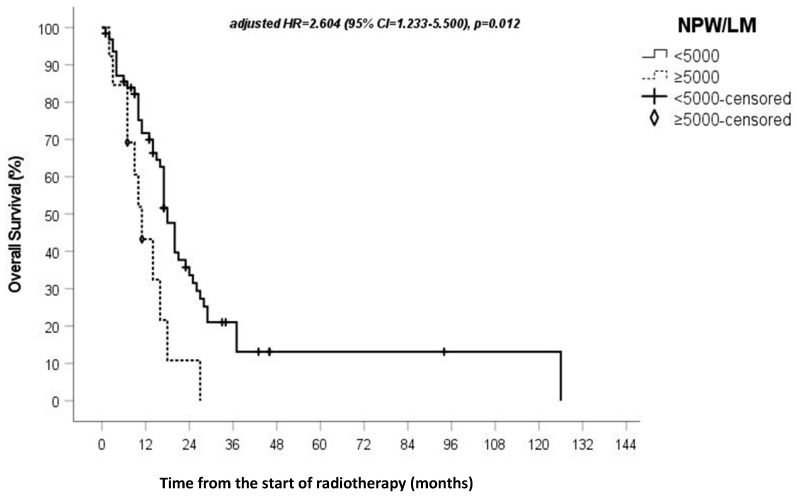
Survival outcomes, based on NPMW/LM value stratification.

**Figure 2 genes-13-01054-f002:**
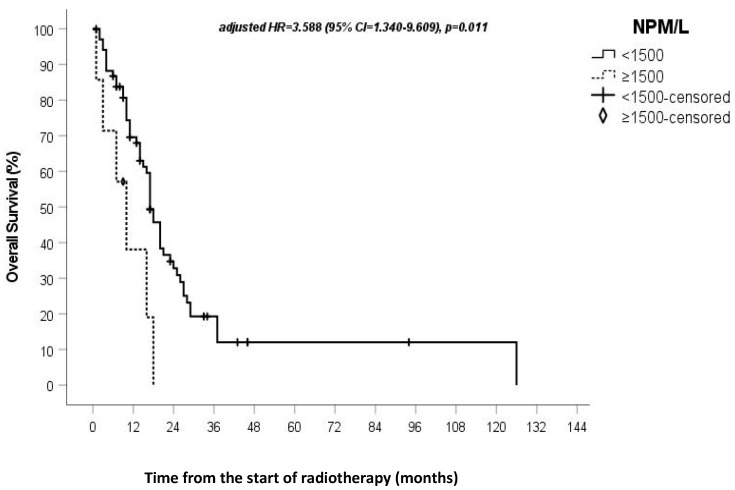
Survival outcomes, based on NPM/L value stratification.

**Table 1 genes-13-01054-t001:** Patients’ clinical characteristics.

Feature	Number
Gender M/F	43/34
Median age (years)	64 (range 26–84)
Second surgery	20 (24.3%)
Median KPS	80
Extent of Surgery: Gross Tumor Removal: 31 (40.2%) Partial Tumor Removal: 46 (59.8%)	
MGMT meth.	23/34 (67.6%)
Blood cells	Mean Value
White blood cells	9.26 (3.55)
Red blood cells	4.70 (0.50)
Hemoglobin	13.9 (1.6)
Platelets	246 (87)
Neutrophils	6.75 (3.29)
Lymphocytes	1.96 (1.39)
Monocytes	0.608 (0.393)
Eosinophils	0.073 (0.082)
Basophils	0.016 (0.013)

KPS: Karnofsky performance status scale, MGMT meth.: methyl-guanine-methyltransferase gene methylation; M: male; F: female.

**Table 2 genes-13-01054-t002:** Univariate and multivariate Cox regression analysis, using the stepwise method, of the OS rate’s predictive categorical factors.

	Univariate Analysis		Multivariate Analysis		
Factor	HR (95% CI)	*p*-Value	RC	HR (95% CI)	*p*-Value
PLR (0) ≤ 250 (1) > 250	2.402 (1.242–4.644)	0.009			0.477
PLRW (0) < 2500 (1) ≥ 2500	2.175 (0.965–4.903)	0.061			0.363
SII (0) < 1200 (1) ≥ 1200	1.848 (1.056–3.234)	0.032			0.924
NPW/L (0) < 7500 (1) ≥ 7500	1.773 (1.045–3.007)	0.034			0.894
NPM/L (0) < 1500 (1) ≥ 1500	2.983 (1.243–7.161)	0.014	1.278	3.588 (1.340–9.609)	0.011
NPMW/L (0) < 8000 (1) ≥ 8000	3.271 (1.269–8.435)	0.014			0.538
NPM/LW (0) < 90 (1) ≥ 90	2.404 (1.148–5.033)	0.020			0.941
NP/LM (0) < 5000 (1) ≥ 5000	2.283 (1.179–4.420)	0.014			0.748
NP/(L + M) (0) < 1300 (1) ≥ 1300	2.263 (1.147–4.463)	0.018			0.736
NPW/LM (0) < 5000 (1) ≥ 5000	2.411 (1.218–4.771)	0.012	0.957	2.604 (1.233–5.500)	0.012
NP/LMW (0) < 1000 (1) ≥ 1000	2.147 (1.036–4.450)	0.040			0.996
NPW/(L + M) (0) < 2500 (1) ≥ 2500	1.852 (0.930–3.689)	0.080			0.860
Surgery (0) GTR (1) STR	2.235 (1.214–4.117)	0.010	0.703	2.020 (1.081–3.775)	0.028

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
