# Peer review of "Old and New Systemic Immune-Inflammation Indexes Are Associated with Overall Survival of Glioblastoma Patients Treated with Radio-Chemotherapy"

_genes, 2022, doi:10.3390/genes13061054_

Round 1

Reviewer 1 Report

A convenient while reliable index to assess GBM patients, such as the reaction to treatment, recurrence, prognosis, etc., is always a hot spot in GBM research. The study introduced the Systemic immunity and inflammation indexes (SI) to the arena. 

  1. More detailed background of the study design is necessary. 
  2. How were the critical values in the SI system defined? 
  3. Is the calculation a mature method? 
  4. Is it a clinically approved system? 
  5. What are the suitable population to use this index system?
  6. Why is the study targeting the IDH 1/2 wild type subgroup of GBM?
  7. The study analyzed the correlation between the SI and the OS. How about other prognosis index, such as the PFS?

Author Response

Dear Editor,

I would like to thank you and your reviewers for your time and expertise, and, mainly, for improving the quality of our article. By following your suggestion, the quality of this document has certainly been improved.

Here is the rebuttal point by point.

Reviewer 1

  • More detailed background of the study design is necessary.

We thank the reviewer for this suggestion.

We have improved the introduction and references.

  • How were the critical values in the SI system defined?

Good comment. We emphasised the method we used to define the critical value of the SIs (The median, 25th, and 75th percentile values were identified for each index).

  • Is the calculation a mature method?
  • Is it a clinically approved system?

As we have specified in the text several times to date, there are no clinical validations and this study is entirely pioneering. Thank you for your comment.

  • What are the suitable population to use this index system?

We agree with the reviewer that the group of patients who may benefit from studying these markers should be better specified. We have supplemented the text.

  • Why is the study targeting the IDH 1/2 wild type subgroup of GBM?

We agree with the reviewer that it should have been better specified why we decided to include only IDH 1/2 wt patients. We have integrated the text.

  • The study analyzed the correlation between the SI and the OS. How about other prognosis index, such as the PFS?

We want to thank the reviewer for this important suggestion. We decided not to analyse PFS because not all patients were evaluated with the same criterion regarding the definition of disease recurrence. The data would not have been reliable.

Reviewer 2

  • As the surgical treatment is the most important factor determining survival and is known for each patient, the presented SI and their impact on OS should be assessed for each type of resection (i.e. Total, partial or biopsy).

We would like to thank the reviewer for this important suggestion. We performed a second statistical analysis by including the impact of surgical radicality in the multivariate analysis.

  • As presented in the discussion part, while the immune modulation by platelets is interesting, authors should determine in their cohort if the higher value of platelets is associated to higher levels of thrombotic events, which are classically described in GBM patients and could correlate to lower survival.

We would like to thank the reviewer for this important suggestion. Several patients in this study started prophylactic anticoagulant therapy immediately after surgery. Therefore, it was impossible to assess a correlation between blood platelet levels and thromboembolic events.

Reviewer 2 Report

Major revision :

  • As the surgical treatment is the most important factor determining survival and is known for each patient, the presented SI and their impact on OS should be assessed for each type of resection (i.e. Total, partial or biopsy).
  • As presented in the discussion part, while the immune modulation by platelets is interesting, authors should determine in their cohort if the higher value of platelets is associated to higher levels of thrombotic events, which are classically described in GBM patients and could correlate to lower survival.

Minor revision :

error line 107 for NPM/L formula

Author Response

(The authors gave the same response as above.)

Round 2

Reviewer 1 Report

The comments have been appropriately addressed by the authors.

Reviewer 2 Report

The authors adapted the manuscript following the type of surgery.